# Medication optimisation in severe mental illness (MEDIATE): protocol for a realist review

Ian Maidment ![ORCID],[1] Geoff Wong,[2] Claire Duddy ![ORCID],[2] Rachel Upthegrove,[3] Sherifat Oduola,[4] Katherine Allen,[5] Simon Jacklin,[6] Jo Howe,[1] Maura MacPhee[7]

[1]College of Health and Life Sciences, Aston University, Birmingham, UK
[2]Nuffield Department of Primary Care Health Sciences, Oxford University, Oxford, UK
[3]Department of Psychiatry, School of Psychology and College of Medical and Dental Sciences, University of Birmingham, Birmingham, UK
[4]Health Service Population Research, University of East Anglia, Norwich, UK
[5]Birmingham and Solihull Mental Health NHS Foundation Trust, Birmingham, UK
[6]School of Pharmacy and Bioengineering, Keele University, Keele, UK
[7]School of Nursing, The University of British Columbia, Vancouver, British Columbia, Canada

**Correspondence to**
Dr Ian Maidment;
i.maidment@aston.ac.uk

## ABSTRACT

**Introduction** Severe mental illness (SMI) is associated with significant morbidity and mortality. People living with SMI often receive complex medication regimens. Optimising these regimens can be challenging. Non-adherence is common and addressing it requires a collaborative approach to decision making. MEDIATE uses a realist approach with extensive engagement with experts-by-experience to make sense of the complexities and identify potential solutions.

Realist research is used to unpack and explain complexity using programme theory/theories that contain causal explanations of outcomes, expressed as context–mechanism–outcome–configurations. The programme theory/theories will enable MEDIATE to address its aim of understanding what works, for whom, in what circumstances, to optimise medication use with people living with SMI.

**Method and analysis** MEDIATE will be conducted over six stages. In stage 1, we will collaborate with our service user/family carer lived experience group (LEG) and practitioner stakeholder group (SG), to determine the focus. In stage 2, we will develop initial programme theories for what needs to be done, by whom, how and why, and in what contexts to optimise medication use. In stage 3, we will develop and run searches to identify secondary data to refine our initial programme theories.

Stage 4 involves selection and appraisal: documents will be screened by title, abstract/keywords and full text against inclusion and exclusion criteria. In stage 5, relevant data will extracted, recorded and coded. Data will be analysed using a realist logic with input from the LEG and SG. Finally, in stage 6, refined programme theories will be developed, identifying causal explanations for key outcomes and the strategies required to change contexts to trigger the key mechanisms that produce these outcomes.

**Ethics and dissemination** Primary data will not be collected, and therefore, ethical approval is not required. MEDIATE will be disseminated via publications, conferences and form the basis for future grant applications.

**PROSPERO registration number** CRD42021280980.

## Strengths and limitations of this study

► Realist methodology is suitable for obtaining detailed understandings of complex areas such as optimising medication use and associated decision making in severe mental illness.
► Including extensive service user and public, practitioner and clinician involvement as part of the methods for the realist review will support the development of robust programme theories/theory and dissemination.
► MEDIATE uses published secondary data, and therefore, is dependent on both the extent and the quality of published data on this topic.

the population, represent 28% of the total disease burden and cost approximately £105 billion every year.[1] Medication is one of the main treatments for severe mental illness (SMI): schizophrenia, bipolar disorder, non-organic psychosis, personality disorder or any other severe and enduring mental illness.[2] Someone living with SMI is estimated to die 15–20 years earlier than the general population, commonly due to the consequences of untreated SMI (eg, suicide) and from physical comorbidities. Side effects of medication can include the worsening of physical comorbidities such as diabetes and cardiovascular disease, confusion and movement disorders, all of which worsen quality of, and shorten, life.[3] Thus, people living with SMI (also called service users) often have complex mental and physical health needs and thus complex medication regimens.

Medication optimisation is 'a person-centred approach to safe and effective medicines use, to ensure the best possible outcomes'.[4] Medication optimisation with people living with SMI is required for the treatments for both mental and physical health needs.[5] Essentially, medication optimisation involves ensuring that people are

## INTRODUCTION

Mental illness has a huge impact. In the UK, mental illness is estimated to affect 25% of

taking medications that are required, and not taking medications that are not required.

Failure to optimise medication can lead to devastating consequences for people living with SMI. Poorly treated mental illness increases the risk of relapse and hospitalisation, and can lead to unemployment, homelessness, disrupted education, substance misuse, physical health problems, and self-harm and excess mortality.[1 2] This can be because of non-adherence and/or underprescribing or overprescribing.[1 2 6 7]

Two important challenges need to be considered when trying to understand medication optimisation namely non-adherence and over-prescribing. Non-adherence occurs in 33%–50% of people with chronic health conditions, with rates in people living with SMI estimated to be up to 50%.[8–10] Non-adherence may lead to worse outcomes and potential relapse, increasing the economic burden on the National Health Service (NHS).[8] Non-adherence is more common in ethnic minority communities.[11] Overprescribing, another important medication issue, is also more common in ethnic minority communities, as are physical illnesses, such as diabetes.[4 7]

The NHS England 5-year plan for mental health identified that people living with SMI often do not receive appropriate support for medication optimisation.[1] A key aspect of medication optimisation is how prescribing decisions are made—what works and what does not work. Collaborative approaches, such as shared-decision making, involving key practitioner groups (pharmacy, medicine and nursing) working with people living with SMI and family carers hold promise for addressing non-adherence and overprescribing practices.[1 8–10 12 13] The potentially coercive nature of treatment for mental health conditions can damage mutual trust between practitioners and people living with SMI, making collaborative approaches more challenging and increasing the risk of medication-related adverse events.[14 15] There is limited research on how prescribing decisions with people with SMI are made, including the use of collaborative approaches.[1 3 11 16]

We have conceptualised medication optimisation with people with SMI as a complex process that has outcomes that vary by context and individuals. Therefore, we have chosen to use a realist review approach to make sense of this complexity. A realist review captures this complexity in a programme theory, which explains the outcomes contained within it using context–mechanism–outcome–configurations (CMOCs).[17] The causal explanations we produce from our analysis will take the form of CMOCs, describing contexts which trigger hidden mechanisms, resulting in outcomes both intended and untended.[18] The overall aim for a realist review is to understand how and which contexts could be manipulated or changed (using different intervention strategies) to trigger mechanisms that achieve desired outcomes and avoid undesired ones.

## METHODS AND ANALYSIS
### Aim
Use published literature and service user, family carer and practitioner engagement to build an understanding of what works, for whom, in what circumstances, to optimise medication use with people living with SMI.

### Objectives
1. To review the literature to understand how and why medication use can be optimised with people living with SMI with a particular focus on the decision-making processes related to medication use.
2. To engage with key stakeholders, including people living with SMI, families, carers and practitioners to identify the key problems and possible solutions.
3. To synthesise the findings from 1 and 2 into realist programme theories for medication optimisation with people living with SMI to identify the key problems and possible solutions.

MEDIATE builds on the approach used in MEMORABLE, which used a realist synthesis to develop intervention(s) to support older people with medication management,[19 20] and in PERISCOPE, which focused on the role of community pharmacy in COVID-19.[21]

To enhance engagement with stakeholder groups (SG), we will hold regular meetings with patient and public involvement (PPI) and practitioner stakeholders. Both groups will be asked to provide advice and feedback on whether our emerging findings and developing programme theories make sense to them. The PPI SG will include people living with SMI, their families and carers—a lived experience group (LEG). Our meetings with the LEG will enable us to better understand their needs, so that their lived experience can inform, initially the realist review, and subsequently, the priorities and design of further research projects. Members of the LEG will primarily be identified from the Birmingham and Solihull Mental Health NHS Foundation Trust (BSMHFT) Lived Experience Action Research Group.

Practitioner SG meetings will be held to understand the challenges for practitioners and their key research priorities. Planned members of the practitioner group include psychiatrists, mental health nurses, occupational therapists, pharmacy staff (both pharmacists and pharmacy technicians) based in primary and secondary care and recovery workers. Members will be identified from practitioner organisations (eg, the College of Mental Health Pharmacy), via BSMHFT (partner trust) and coapplicant professional links.

Over six stages, MEDIATE will use secondary data with input from key stakeholders to develop refined programme theories. Realist And Meta-narrative Evidence Syntheses: Evolving Standards (https://www.ramesesproject.org/) guidance will be followed.

### Stage 1: focusing the review
To ensure that our review will focus on the issues related to medication optimisation that are important to people

living with SMI, we will hold initial consultative meetings with the LEG and SG. During these meetings, we will explain our plans, to obtain their views on our overall approach, and to get their feedback and advice on key medication optimisation issues to include in our review. Each initial meeting will last approximately 90 min; prior to the meetings participants will be emailed a summary of MEDIATE, our plans and the overall roles of the group including how they can contribute.

## Stage 2: developing initial programme theories

We will develop a 'rough and ready' explanation for what needs to be done, by whom, how and why, and in what contexts, to optimise medication for people living with SMI. The initial programme theories will be refined as the review progresses. Based on our experience in MEMO-RABLE, we may need one or more programme theories for different settings (eg, community vs supported) and different populations, including ethnic minorities. The initial programme theories will be developed using the content expertise of the project team and a preliminary review of the medication optimisation literature. These initial programme theories will then be presented to the LEG and SG groups for their feedback and ongoing refinement.

## Stage 3: developing search strategy

Further programme theory refinement will use secondary data from the academic and grey literature. We will use iterative literature searches with different search term concepts and permutations at predetermined milestones to capture the most relevant data.[22 23] Our information specialist (CD) will develop, refine and run the searches for this project, seeking input from the wider project team, and the LEG and SG, as needed. Once we develop the initial programme theory/theories we will then be able to develop our initial search strategy.

The proposed initial sampling frame, to be used as the basis for comprehensive literature search strategies, is as follows:
► Context: adults living with SMI on medication.
► Intervention or phenomenon: any intervention to optimise medication usage; people living with SMI, family carers' and practitioners' experiences of managing and using medication.
► Mechanisms: triggered by the intervention, to be identified from the programme theories.
► Outcomes: quality of life, adherence, adverse events, disease symptoms, economic. Unanticipated or unintended outcomes, and outcomes considered important by our LEG and SG may also be identified and included later.

Based on discussions with our information specialist (CD), sources will include: MEDLINE/PubMed, Embase, Scopus, Web of Science (Core Collection Indexes), the Cochrane Library, CINAHL, PsycINFO, Sociological Abstracts and Google Scholar. Additional grey literature will be sought by searching EThOS (British Library Electronic Theses Online), ProQuest Dissertations and Theses, OpenGrey (System for Information on Grey Literature in Europe), the King's Fund Library Database, NHS Evidence and the websites of relevant charities/user groups/professional bodies. To identify further grey literature, such as unpublished service evaluations, we will use professional networks (eg, College of Mental Health Pharmacy, the British Association of Psychopharmacology) and relevant NHS organisations.

Where necessary, we will use 'cluster searching' techniques to identify additional papers that might add to the conceptual and contextual richness of studies initially identified within the sampling frame constructed through conventional topic-based searching.[23] For example, we will aim to identify 'sibling' (ie, directly linked outputs from a single study) and 'kinship' (ie, associated papers with a shared contextual or conceptual pedigree) papers and reports.[23] We will also conduct forward and backward citation searches, using Google Scholar and Web of Science, to identify further related papers from the wider literature, and approach our LEG and SG for recommendations for potentially relevant documents. Searching will continue until sufficient data is found ('theoretical saturation') to conclude that the refined programme theories are sufficiently coherent and plausible.[22]

If the volume of retrieved literature proves unmanageable, we will employ a variety of appropriate sampling strategies (eg, theoretical sampling, maximum variation sampling, extreme case sampling) to ensure that we have sufficient focused but relevant data for programme theory development.[24]

## Stage 4: selection and appraisal

Selection and appraisal will be a two-step process:
1. Potentially relevant documents will initially be screened by title, abstract and keywords by the research associate (JH), against inclusion and exclusion criteria which we will develop based on the context–intervention–mechanism–outcome framework above, with input from the LEG and SG.[22] A 10% random sample will be checked (by MM) for any systematic errors (any disagreements will be resolved with the input of IM).
2. The full texts of this set of documents will be obtained and screened by the research associate against inclusion and exclusion criteria. We will use a similar checking process as for step 1 above. The research associate (JH) will read the full text of all the documents that have been included. Documents will be selected for inclusion based on:

### Relevance
Are sections of text within this document relevant to programme theories development?

### Rigour
Are these data sufficiently trustworthy to warrant making changes to the programme theories?

To illustrate how we will operationalise rigour, if relevant data have been generated using a qualitative approach, then the trustworthiness of the data would be considered to be greater if the data was (eg) triangulated with people living with SMI, family (informal) carers and clinicians interviewed. Documents may still be included even if judged to be of limited rigour, as we will also make an overall assessment of rigour at the level of the programme theory.

## Stage 5: data extraction and analysis/synthesis

The research associate (JH) will upload full texts of included papers into NVivo (a qualitative data analysis software tool). Relevant sections of texts that are pertinent to medication optimisation contexts, mechanisms and their relationships to outcomes will be coded in NVivo. This coding will be inductive (codes created to categorise data reported in included studies), deductive (codes created in advance of data extraction and analysis as informed by the initial programme theory) and retroductive (codes created based on an interpretation of data to infer what the hidden causal mechanisms might be for outcomes). The characteristics of the documents will be extracted separately into an Excel spreadsheet. Each new element of relevant data will be used to refine the programme theory, and as the theory is refined, included studies will be re-scrutinised to search for data relevant to the revised theory that may have been missed initially.[22 25]

Data analysis will use a realist logic of analysis to make sense of the initial programme theories. The research associate (JH) will undertake this step with support from the project team and input from the LEG and SG. We will use a series of questions to support our analysis and synthesis of data (in addition to relevance and rigour)[26]:

▶ Interpretation of meaning: if relevant and trustworthy, do the contents of an included document provide data that may be interpreted as functioning as context, mechanism or outcome?
▶ Interpretations and judgements about CMOCs. For example, what is the CMOC (partial or complete) for the data that has been interpreted as functioning as context, mechanism or outcome?
▶ Interpretations and judgements about programme theory. For example, how does this particular (full or partial) CMOC relate to the programme theory or theories? Within this same document, are there data, which informs how the CMOC relates to the programme theory or theories?

Data to inform our interpretation of the relationships between contexts, mechanisms and outcomes will be sought across documents, because not all parts of the configurations will always be articulated in the same document. Interpretive cross-case comparison will be used to understand and explain how and why observed outcomes have occurred, for example, by comparing and contrasting settings where people living with SMI are either more or less actively involved in decision making during medication optimisation. When working through the questions set out above, where appropriate, we will use the following forms of reasoning to make sense of the data: juxtaposition of data, reconciling of data, adjudication of data and consolidation of data.[26]

## Stage 6: programme theory development

Refined programme theories for medication optimisation with people living with SMI will be based on:
▶ The key outcomes that are important to people living with SMI, family carers and practitioners.
▶ Using the data to identify key mechanisms that need to be 'triggered' for desired outcomes from medication optimisation.
▶ Identifying contexts related to these key mechanisms and the strategies required to change the contexts to trigger the key mechanisms for desired outcomes.

We will hold separate meetings with the LEG and SG to review the programme theories and provide their perspective on the most promising intervention strategies to pursue with future programme development. In addition, we will consider holding some joint meetings if members of the LEG recommend this approach.

## Patient and public involvement

Patient and public involvement is central throughout MEDIATE and led by the LEG containing approximately 8–12 service users and family carers, who will advise on the conduct of the research. The development of the research question was informed by service users' priorities, experience and preferences obtained from focus groups involving approximately 25 service users/carers of mixed genders and ethnicities linked with the SURE-SEARCH Mental Health Network and the BSMHFT Lived Experience Action Research Group. These groups identified the importance of the issue, that often their views are not listened to and the coercive nature of treatment in the context of unequal power relationships. Service user and family carer representatives advised on the design in particular the outline focus of the realist review.

Service users will be equal partners in dissemination. They will advise the team on the most appropriate ways to disseminate the results of MEDIATE to service users and family carers.

## Ethics and dissemination

Primary data will not be collected and therefore ethical approval is not required. The results will be disseminated via peer-reviewed publications and conference presentations. The realist review will form the basis for future grant applications and provide the required knowledge to develop a complex intervention to optimise medication use with people living with SMI, specifically: clear rationales for the most promising intervention strategies, and appropriate outcome measures to evaluate an intervention(s), including the use of realist data to inform Health Economic Modelling.[27]

Aspects of a complex intervention, that further research could investigate could include:

- ► Promotion health literacy.
- ► Peer-to-peer support, including, for example, how pharmacy could support this (planned to be led by people living with SMI).
- ► The role of family carers.
- ► The role of social care and third sector organisations.
- ► Practical steps including alerts, reminders, adherence aids, prescription timing.

**Contributors** IM is the guarantor of the review and lead investigator on MEDIATE. IM, GW, MM and CD contributed equally to designing the review and drafting the methodological details of MEDIATE. KA advised on the PPI aspects of MEDIATE. RU, SO, MM, GW and SJ contributed to the design of MEDIATE participated in study design and knowledge translation. SO, RU, MM and GW on key clinical aspects and SJ on decision-making processes. All authors contributed to drafting the protocol, have read and approved the final manuscript. We acknowledge the contribution of service user advisors via the BSMHFT Lived Experience Action Research (LEAR) group and SURESEARCH Mental Health Network.

**Funding** This study/project is funded by the National Institute for Health Research (NIHR; Programme Development Grant: 203683).

**Disclaimer** The views expressed are those of the author(s) and not necessarily those of the NIHR or the Department of Health and Social Care.

**Competing interests** GW is deputy chair of the UK's National Institute of Health Research Health Technology Assessment Prioritisation Committee: Integrated Community Health and Social Care Panel (A) and a member of Methods Group (A).

**Patient and public involvement** Patients and/or the public were involved in the design, or conduct, or reporting, or dissemination plans of this research. Refer to the Methods section for further details.

**Patient consent for publication** Not applicable.

**Provenance and peer review** Not commissioned; externally peer reviewed.

**ORCID iDs**
Ian Maidment http://orcid.org/0000-0003-4152-9704
Claire Duddy http://orcid.org/0000-0002-7083-6589

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
