## [Reviewer comments · BMJ Open]

ARTICLE DETAILS

TITLE (PROVISIONAL)	MEDication optimisATIion in severE mental illness (MEDIATE): Protocol for a Realist Review
AUTHORS	Maidment, Ian; Wong, Geoff; Duddy, Claire; Upthegrove, Rachel; Oduola, Sherifat; Allen, Katherine; Jacklin, Simon; Howe, Jo; MacPhee, Maura

VERSION 1 – REVIEW

REVIEWER	Garfield, Sara Imperial College Healthcare NHS Trust
REVIEW RETURNED	03-Dec-2021

GENERAL COMMENTS	The proposed work will make an extremely valuable contribution to the literature and practice. The background is very well written. I have a few fairly minor suggestions: 1. BMJ Open readers may not be familiar with realist review and understand terms such as programme theory, context and mechanism. I recommend adding some more background to the realist review approach.2. The way the objectives are written makes the study sound like an empirical study, rather than a review. It would be helpful to modify these to make this clearer.3. There are quite a few abbreviations used which makes the paper more difficult to follow. Try to cut these down if possible. Please also define all study acronyms used.4. I found the abstract very difficult to follow. I appreciate that it may be difficult to summarise a complex methodology in the word count limits. My suggestion would be to include less details of the 6 stages and more overall explanation.
---

VERSION 1 – AUTHOR RESPONSE

Comments to the Author:

The proposed work will make an extremely valuable contribution to the literature and practice. The background is very well written. I have a few fairly minor suggestions:

1. BMJ Open readers may not be familiar with realist review and understand terms such as programme theory, context and mechanism. I recommend adding some more background to the realist review approach.

As requested, we have added a summary on realist reviews and explain terms such as programme theory, context and mechanism both in the abstract and the main text (see page 6).

2. The way the objectives are written makes the study sound like an empirical study, rather than a review. It would be helpful to modify these to make this clearer.

As requested, we have reviewed and modified the objectives to make it clearer that MEDIATE involves reviewing the literature.

3. There are quite a few abbreviations used which makes the paper more difficult to follow. Try to cut these down if possible. Please also define all study acronyms used.

We have significantly reduced the number of abbreviations (removed many including e.g., BAP, CMHP, CIMO, BAP). For the remaining abbreviations we have put a study acronym table on page 15.

4. I found the abstract very difficult to follow. I appreciate that it may be difficult to summarise a complex methodology in the word count limits. My suggestion would be to include less details of the 6 stages and more overall explanation.

As requested, we have included less detail in the stages and added more explanation.

We have now appointed a Research Associate who has significantly contributed to revising and editing this re-drafted protocol. We have therefore added Jo Howe as a co-author.

We look forward to hearing from you at your earliest convenience.

VERSION 2 – REVIEW

REVIEWER	Garfield, Sara Imperial College Healthcare NHS Trust
REVIEW RETURNED	23-Dec-2021
GENERAL COMMENTS	Thank-you for making the requested revisions